# The Impact of COVID-19 on Physical (In)Activity Behavior in 10 Arab Countries

**DOI:** 10.3390/ijerph191710832

**Published:** 2022-08-31

**Authors:** Haleama Al Sabbah, Zainab Taha, Radwan Qasrawi, Enas A. Assaf, Leila Cheikh Ismail, Ayesha S. Al Dhaheri, Maha Hoteit, Ayoub Al-Jawaldeh, Reema Tayyem, Hiba Bawadi, Majid AlKhalaf, Khlood Bookari, Iman Kamel, Somaia Dashti, Sabika Allehdan, Tariq A. Alalwan, Fadwa Hammouh, Mostafa I. Waly, Diala Abu Al-Halawa, Rania Mansour, Allam Abu Farha

**Affiliations:** 1Department of Health Sciences, Zayed University, Dubai P.O. Box 19282, United Arab Emirates; 2Department of Health Sciences, Zayed University, Abu Dhabi P.O. Box 144534, United Arab Emirates; 3Department of Computer Science, Al-Quds University, Jerusalem 20002, Palestine; 4Department of Computer Engineering, Istinye University, Istanbul 34010, Turkey; 5Faculty of Nursing, Applied Science Private University, Amman 11931, Jordan; 6Department of Clinical Nutrition and Dietetics, University of Sharjah, Sharjah 27272, United Arab Emirates; 7Nuffield Department of Women’s & Reproductive Health, University of Oxford, Oxford OX1 2JD, UK; 8Department of Nutrition and Health, College of Medicine and Health Sciences, United Arab Emirates University, Al Ain P.O. Box 15551, United Arab Emirates; 9Faculty of Public Health, Lebanese University, Beirut P.O. Box 11-0236, Lebanon; 10PHENOL Research Group (Public Health Nutrition Program Lebanon), Faculty of Public Health, Lebanese University, Beirut P.O. Box 11-0236, Lebanon; 11Lebanese University Nutrition Surveillance Center (LUNSC), Lebanese Food Drugs and Chemical Administrations, Lebanese University, Beirut P.O. Box 11-0236, Lebanon; 12World Health Organization Regional Office for the Eastern Mediterranean, Cairo 11884, Egypt; 13Department of Human Nutrition, College of Health Sciences, Qatar University, Doha 2713, Qatar; 14Department of Nutrition and Food Technology, School of Agriculture, The University of Jordan, Amman 11942, Jordan; 15National Nutrition Committee, Saudi Food and Drug Authority, Riyadh 11451, Saudi Arabia; 16Department of Clinical Nutrition, Faculty of Applied Medical Sciences, Taibah University, Madinah 42353, Saudi Arabia; 17National Research Centre, Cairo 11884, Egypt; 18Public Authority for Applied Education and Training, Kuwait City 13092, Kuwait; 19Department of Biology, College of Science, University of Bahrain, Sakhir P.O. Box 32038, Bahrain; 20Department of Nutrition and Dietetics, Faculty of Health Sciences, American University of Madaba, Amman 11821, Jordan; 21Food Science and Nutrition Department, College of Agricultural and Marine Sciences, Sultan Qaboos University, Muscat 123, Oman; 22Faculty of Medicine, Al-Quds University, Jerusalem 20002, Palestine; 23Doha Institute for Graduate Studies, Doha P.O. Box 200592, Qatar; 24College of Business and Economics, Qatar University, Doha 2713, Qatar

**Keywords:** COVID-19, coronavirus, pandemic, physical activity, quarantine, lockdown, adults

## Abstract

Insufficient physical activity is considered a strong risk factor associated with non-communicable diseases. This study aimed to assess the impact of COVID-19 on physical (in)activity behavior in 10 Arab countries before and during the lockdown. A cross-sectional study using a validated online survey was launched originally in 38 different countries. The Eastern Mediterranean regional data related to the 10 Arabic countries that participated in the survey were selected for analysis in this study. A total of 12,433 participants were included in this analysis. The mean age of the participants was 30.3 (SD, 11.7) years. Descriptive and regression analyses were conducted to examine the associations between physical activity levels and the participants’ sociodemographic characteristics, watching TV, screen time, and computer usage. Physical activity levels decreased significantly during the lockdown. Participants’ country of origin, gender, and education were associated with physical activity before and during the lockdown (*p* < 0.050). Older age, watching TV, and using computers had a negative effect on physical activity before and during the lockdown (*p* < 0.050). Strategies to improve physical activity and minimize sedentary behavior should be implemented, as well as to reduce unhealthy levels of inactive time, especially during times of crisis. Further research on the influence of a lack of physical activity on overall health status, as well as on the COVID-19 disease effect is recommended.

## 1. Introduction

The World Health Organization (WHO) declared COVID-19 a global pandemic due to the worldwide spread of the novel coronavirus disease (COVID-19). Under the circumstances defined by the WHO and other international and national organizations as pandemics, different countries have formulated preventive and precautionary measures recommended by governments to enforce control of the situation [1]. However, most of these governmental measures have resulted in quarantine and lockdowns to contain the spread of the virus and protect vulnerable populations [2].

Among the various measures recommended by the WHO and followed in different countries, including the Middle East, is nationwide lockdown and quarantine, which requires people to stay at home. Most health ministries in the Middle East have recognized COVID-19 as a major threat to civilians and have highlighted the importance of following the guidelines and measures set out by the WHO [3]. Following the WHO and governments in many countries demanded that people practice physical distancing and only depart their homes for the necessary reasons [1]. However, these changes were accompanied by a more sedentary lifestyle due to increased screen time [4], smart working and the resultant abolition of walking or public transport to work, and fewer opportunities for practicing physical activity (PA) [5].

Although the strict preventive measures recommended by the WHO are crucial for protecting people’s health and lives, they implicated some effects that cannot be ignored. Considering these public health measures, people would be expected to face some changes in their physical activity behavior. For instance, the closure of the commercial gym facilities, playgrounds, and city parks, would require them to consider performing indoor physical activity practices. While the health authorities take public health measures in these countries, mainly to ascertain that people stay as healthy and safe as possible, the unintended results may be a reduction in PA and an increase in sedentary behavior. This will expose the population to an increased risk of chronic health conditions. As people are confined and work from home, their PA is largely restricted [6]. This is because of the unfamiliar nature of indoors for performing physical activities or the over-reliance on outdoor facilities, such as commercial gyms, outdoor calisthenics parks or running trail ways, and equipment to practice regular PA [7,8].

The effect of confinement on people during quarantine should be seriously considered when thinking about the prevalence of obesity globally, as well as in the Middle East [9,10]. Inactivity and sedentary behavior have wide-ranging adverse impacts on human health at the muscular, cardiovascular, metabolic, and endocrine levels, and affect psychological well-being [11,12]. The Middle East/North Africa (MENA) region has been reported as having the highest concentration of non-communicable diseases worldwide [13]. The region has also been declared to have the second-highest prevalence of diabetes in the world (10.8%) [14] and is recording a rapid increase in obesity [15,16]. Insufficient PA and a sedentary lifestyle are considered strong risk factors associated with obesity and other NCDs, leading to premature mortality [17]. Accordingly, PA has been well-documented as an important factor in the control and prevention of obesity [18].

Several studies have investigated the effects of the COVID-19 pandemic and the measures taken and implemented in different countries on PA; most of the results showed a significant decrease in the amount of time spent on physical activity levels during the COVID-19 lockdown than before the lockdown period such as in the UAE, Saudi Arabia [19]. The results of extensive research data collected by an international survey revealed an increase in daily sitting time during the pandemic restrictions, ranging from 5 to 8 h per day [20,21]. A study was conducted in the Greater Middle East region, including the 10 countries investigated in the current study, Bahrain, Egypt, Jordan, Kuwait, Lebanon, Oman, Palestine, Qatar, Saudi Arabia, and the United Arab Emirates. The results revealed that the lockdown due to the COVID-19 pandemic resulted in several lifestyle changes, including physical inactivity [22]. This should be taken seriously when considering the low level of PA and the high prevalence of obesity at both the international and national levels [9,10]. Accordingly, health authorities in the Middle East should take action to address this situation. As with COVID-19, the level of PA is going to be even worse, which will affect people’s health in relation to obesity and the risk of other chronic diseases.

External motivation may be required for people to be physically active and to regularly engage in exercise. Motivation can be provided through physical or social–mental stimuli. Unfortunately, during the COVID-19 confinement period, people might have lost motivation due to stress and fear of the pandemic itself [23]. Studies have investigated the effects of confinement and quarantine on people’s health in terms of psychological and mental effects such as stress, confusion, and anger [24,25].

It is logical to consider the effect of the COVID-19 pandemic on PA, not only in terms of confinement and quarantine but also in the associated financial issues. Job loss due to COVID-19 has impacted people’s lives and readiness to engage in PA. Under these conditions, people may be less motivated to consider fitness programs available on different websites, thereby promoting and encouraging indoor exercise routines during the pandemic. As per the available studies conducted to assess the effect of the COVID-19 pandemic on PA and investigate its factors, the results are still insufficient to draw strong conclusions that might help formulate policies and strategies on this important issue [26,27]. Our study analyzed data for a large sample collected from 10 Arab countries with an objective that measured physical (in)activity using repeated variables before and during the COVID-19 lockdown. The aim of this study is to assess the impact of COVID-19 on physical levels before and during the lockdown, and to identify the factors affecting people’s PA in the Middle East. We hypothesize that COVID-19 would negatively impact the physical (in)activity behaviors in the 10 Arab countries.

## 2. Materials and Methods

### 2.1. Study Design and Settings

A retrospective observational study with a cross-sectional design using an online validated survey, was launched originally in 38 different countries, and information collected from 37,207 participants. The Eastern Mediterranean regional data related to 10 Arabic countries (United Arab Emirates, Lebanon, Bahrain, Egypt, Jordan, Kuwait, Oman, Qatar, Saudi Arabia, and Palestine) were selected for the sake of analysis in this study. The international study protocol was approved by the Ethics Committee for the Social Sciences and Humanities of the University of Antwerp (ref. no: SHW_20_46). For the 10 Arab countries, the study protocol was revised and adapted to suit the culture of Arabs and then approved by Zayed university ethical committee (ref. no: ZU20_098_F). The survey was kept open between 17 April and 25 June 2020 and consisted of multiple blocks of information. The details of the study methodology have been described elsewhere [28,29].

### 2.2. Population and Sampling

Participants included in this study were of an age exceeding 18 years old, of both genders, and residing in any of the participating Arabic countries. Children and adolescents <18 years old were excluded for this study. Convenience snowball sampling was used to recruit respondents, and advertisements for the survey were conducted using different social network platforms, as well as the research team’s academic networks. First of all, the international research team created and shared multiple social media banners on sites such as Facebook, Twitter, Instagram, and LinkedIn in both private and public online groups. Furthermore, an international press release was distributed to the spokespersons of the research teams in each country, who were able to distribute it to their local press. In addition, the international news agency Reuters produced a video article about the CoronaCookingSurvey, which was distributed to several international press organizations. The CoronaCookingSurvey was mentioned in newspapers, on the radio, or on news websites in 24 different countries, so it was helpful to share the link to a global web page where participants could find the right survey link for their own country [30].

### 2.3. Measurement and Data Collection Procedure

The survey was a self-administered, validated, online questionnaire conducted using the survey software Qualtrics. It took approximately 20 min to complete. All countries used the same survey structure as a starting point. Extra questions were added at the end of the questionnaire, and some questions were adapted to better reflect the sociocultural context (e.g., alcohol use in Arab countries). An ad hoc questionnaire, created for the present study, took into account items from other questionnaires such as Health Behavior in School-aged Children (HBSC) (http://www.hbsc.org/methods/, accessed on 15 July 2022). Moreover, to make sure the questionnaire is valid and reliable, the survey questionnaire and the additional questions were reviewed and discussed by all principal researchers of the 10 Arab countries that participated in this study via several ZOOM meetings and WhatsApp groups created for this research project.

Physical activity level was measured using two questions: (a) Over the past 7 days, on how many days were you physically active for a total of at least 60 min per day? (b) Over a typical or usual week, before the lockdown, on how many days were you physically active for a total of at least 60 min per day? Response categories were: 0 days, 1, 2, etc., up to 7 days. The responses for each question were categorized into: (1) physically active if the response was >5 days a week, (2) moderate activity 3–4 days a week, low activity 1–2 days a week, and inactive if the response was 0 days a week [31].

The complete questionnaire consisted of six main parts in addition to the extra questions specific to Arab countries: (1) profiling questions, (2) lockdown and consequences, (3) general food behavior, (4) grocery shopping, (5) cooking and baking, and (6) eating behavior. The last part of the questionnaire used in the 10 Arab countries included the extra questions about dieting, physical (in)activity, body image, smoking behaviors, and self-reported weight and height. The survey questions were available in the native Arabic language as well as English, extending choices for the respondents.

In addition to the International ethical approval and the approval from the Ethics Advisory Committee on Social and Human Sciences at the University of Antwerp (Ref. No.: SHW_19_44), ethical approval was sought on 26 April 2020 from Zayed university’s Ethics Committee before data collection (Ref. No.: ZU20_098_F). Before moving on to the survey question, respondents had to read the following: (1) what the study is about (objective), (2) who can participate, (3) what are the rights and responsibilities, and (4) the contact details of the principal investigators. Finally, they were asked to provide their fully informed consent and say whether they were above 18 years, immediately after the survey’s welcome page (https://osf.io/r9n25/, accessed on 15 July 2022).

### 2.4. Data Analysis

A total of 12,433 participants from 10 Arab countries who reported their PA behavior before and during the COVID-19 confinement were included. Changes in PA during the COVID-19 confinement were analyzed and compared according to several factors, such as gender, age, education, working status, screen watching, and country. Paired sample *t*-test analysis was used for comparing the means of PA groups before and during the lockdown. Multiple regression analysis was conducted to examine the relationship between PA and participants’ countries. Linear regression analysis was used for the continuous variables of PA before and during the COVID-19 lockdown. A one-way repeated-measures analysis of variance (ANOVA) was conducted to evaluate the null hypothesis (there is no change in participants’ physical activity levels before and during the COVID-19 lockdown) [32,33]. The significant level was set at *p* < 0.05. Statistical analysis was conducted on IBM SPSS Statistics, Version 25 (IBM, Armonk, NY, USA).

## 3. Results

A total of 12,433 people from 10 Arab countries fully completed an online survey. Respondents were 80% females and 20% male (9951 and 2482, respectively). The mean age of the participant was 30.3 (SD, 11.7) years. The respondents were categorized into five age groups, as shown in Table 1. Approximately 50% of the participants were aged between 20 and 29 years.

Table 2 presents the percentage, mean and standard deviation (M ± SD) of physical activity before and during the lockdown by country. The results indicated that the percentage of physically inactive participants increased in all countries. The highest increases were found in Egypt, Kuwait, Qatar, and the Emirates (29.9 to 43%, 24.6 to 35.2%, 23.2 to 33.8%, 23.3 to 31.9%, respectively). Other countries reported an average increase of 6%. Furthermore, the results in Table 2 reported −5.5% and −4.3% average percentage decrease in moderate and highly active participants, respectively. The highest reduction in moderate PA was in Qatar (−13.3%), while the highest decline in highly active participants was in Egypt (−7.3%).

Table 3 shows the univariate analysis of PA with participants’ demographic characteristics. The results indicated that the percentage of people who became physically inactive was higher among males than among females (23.8 to 32.3% and 30.7 to 37.5%, respectively). Moderate and highly active participants reported an average percentage decrease of approximately −4% in males and −3% in females. The results of PA by education showed that the bachelor’s and graduate groups had the highest increase in inactive PA (30 to 37.3% and 23.5 to 33.3%, respectively). The graduate education level group reported the highest decrease in moderate PA (36.8 to 29.6%). Those with less than high school and bachelor’s education reported a higher reduction in the highly active category (14.2 to 10.1% and 12.9 to 8.8%, respectively). The results indicated that the working people before the lockdown reported a higher increase in the inactive category than the non-working people before the lockdown (28.8 to 36.3% and 29.9 to 36.6%, respectively). The working group reported a higher increase in the inactive category than the non-working group (27.9 to 35.6% and 29.9 to 36.9%, respectively). Furthermore, the results indicated that the average decrease in moderate and highly active groups in work groups before and during the lockdown was ~4%. The results indicated that the highest increases in the inactive level were found in the 30–39, 40–49, and 50+ years age groups (31.3 to 39.2%, 33.8 to 41.4%, and 32.2 to 40.9%, respectively). The highest decreases in the moderate level were found in the 30–39 and 50+ years age groups (31.9 to 25.2%; 30.4 to 24.7%). Furthermore, the young ages reported the highest decrease in the highly active level 18–19 and 20–29 age groups (14.9–9.7%; 13.9–9.2%), respectively.

Table 4 shows the percentage of PA before and during the lockdown, along with the time spent watching TV and using the computer. The results of the univariate analysis indicated a significant effect of the time spent watching TV on PA before and during the lockdown (F (3, 12430) = 9.6, *p* < 0.001), F (3, 12430) = 8.7, *p* < 0.001)). The inactive group, with the time spent on watching TV, reported an average increase of 6.4 % during the lockdown compared to the same group before the lockdown. The moderate and highly active groups reported a 4% decrease during the lockdown compared to the same group before the lockdown. The moderate and highly active groups reported a 4% decrease during the lockdown compared to the same group before the lockdown. Furthermore, the results in Table 4 show a significant effect of using a computer on physical activities before and during the lockdown (F (3, 12430) = 14.5, *p* < 0.001, F (3, 12430) = 8.4, *p* < 0.001). The inactive group reported an average increase of 7% in time spent using the computer during the lockdown compared to the same group before the lockdown. The moderate and highly active groups reported a 4% decrease in the time spent using the computer during the lockdown compared to the same group before the lockdown.

Multiple regression analysis was conducted to examine the relationship between PA and the participants’ country. Table 5 summarizes the regression results. It can be seen from the results that the participants’ gender and education have a positive association with PA before and during the lockdown (*p* < 0.05). The age group, watching TV, and using computers had a negative effect on PA before and during the lockdown (*p* < 0.05), indicating that older participants with higher ages, time spent on watching TV and using a computer were expected to have lower physical activities. Furthermore, the results indicated a significant association between PA and gender, education, age, watching TV, and computer use.

The result of the ANOVA indicated a significant difference found between the two groups. Wilks’ Lambda = 0.96, F (1, 12432) = 515.8, *p* < 0.001, η^2^ = 0.04. Thus, there is significant evidence to reject the null hypothesis. Follow-up comparisons indicated that the pairwise difference was significant, *p* < 0.001. There was a significant increase in scores between the two groups, suggesting that COVID-19 lockdown affects participants’ physical activity levels. A paired sample *t*-test was conducted for examining the effect of COVID-19 on physical activity levels. The *t*-test compared the means for the two groups (before and during the lockdown). Results in Table 6 indicated that PA scores before and during the lockdown were positively correlated, and there was a significant difference between the two groups (*t*_12433_ = 24.1, *p* < 0.001). On average, PA scores before lockdown were 0.193 higher than PA scores during lockdown (95% (0.178–0.209)). The results indicated a significant average difference between participants’ work before and during the lockdown scores (*t*_12433_ = 31.6, *p* < 0.001). On average, the work before was 0.086 points higher than during the lockdown (95% (0.08–0.091)). The results indicated a significant average difference between time spent on watching TV before and during lockdown scores (*t*_12433_ = −73.7, *p* < 0.001). On average, the time spent watching TV before was −0.548 points lower than during the lockdown (95% (−0.562–−0.533)). The results indicated that there was a significant average difference between time spent on using a computer before and during lockdown scores (*t*_12433_ = −66.9, *p* < 0.001). On average, the time spent on using a computer before was −0.489 points lower than during the lockdown (95% (−0.504–−0.475)).

Table 7 shows the results of the linear regression analysis of the continuous variable of PA before and during the COVID-19 lockdown. The results revealed a decrease in PA during the lockdown in most Arab countries. Furthermore, significant differences were found between some of the Arab countries and physical activities during the lockdown, including Egypt, Jordan, Lebanon, Saudi Arabia, and Palestine (*β* = −0.144, *p* = 0.000; *β* = −0.057, *p* = 0.004; *β* = −0.070, *p* = 0.001; *β* = −0.078, *p* = 0.000; *β* = −0.139, *p* = 0.000).

## 4. Discussion

To our knowledge, this is the first study to assess the impact of the COVID-19 pandemic and associated factors on PA behavior across a range of Arab nations. The findings of the study support the hypothesis that COVID-19 had negatively impacted the physical activity participation in the 10 Arab countries. The results show that all countries had increased percentages of physically inactive participants during the lockdown. The highest increases were found in Egypt, Kuwait, Qatar, and the UAE, while other countries similarly reported a minor average increase of 6%. The highest decrease in highly active participants was in Egypt (−7.3%). This significant prevalence of lower physical activity was also reported by other researchers in the UAE (30% decreased PA), Lebanon (41% no PA), and Kuwait (33.1% decreased PA) during lockdown [34,35,36].

These results are in line with the majority of studies, which found that PA decreased and inactivity rose during the COVID-19 pandemic lockdown [37,38,39]. Several studies have shown that the PA during lockdown decreased compared with pre-lockdown, despite the efforts of different health organizations and exercise guidelines that encourage people to stay active during the pandemic and in self-quarantine [40,41,42].

On the other hand, some studies have reported no significant changes in the level of PA before and after the lockdown [43,44,45,46]. Differences between countries could be related to the governmental measures used in each country, which showed some differences. In Egypt, for instance, young people (20–29), who are the majority target age group, were active in terms of using sport facilities, walking, working, and using transportation, among other activities. Sport facilities were shut down during the lockdown, forcing students to study at home. For the government and private sectors, however, work was split into alternating days, with each person working 2–3 days per week. These actions greatly reduce activity in Egypt [47]. The findings showed that working people before the lockdown reported a higher increase in the inactive category than the non-working people before the lockdown. However, other researchers have found that PA is rising among working adults [44].

Our findings showed differences in PA with respect to gender; men were more likely than women to report being physically inactive. Similar results were reported by other researchers, who found that women compared to men showed a lower tendency to reduce PA levels during the lockdown [48]. According to the findings, the highly active level decreased most rapidly in young age groups. Similar results were reported in other studies, which found that PA levels decreased in the younger age and student groups [49,50].

About 50% of participants were in the group (20–29 years). Data collection modes may affect response bias. Previous studies have demonstrated that the mode of data collecting may affect the responses of demographic subgroups. For instance, numerous studies have revealed that younger people prefer to answer via the web and email, but elderly people prefer non-web ways [51]. Therefore, it is important to take into account when interpreting the findings because 50% of the participants in the current study are between the ages of 20 and 29. The findings of this study that pinpoint the elements that influence PA behavior, such as gender, age, and employment, would be extremely significant from the perspective of public health. The majority of research investigations on the effects of COVID-19 on PA revealed increases in inactivity during the lockdown. Given that many people worked from home, which increased sedentary time and screen time, this result should not be interpreted as being out of the ordinary [52,53].

Recreational screen time such as watching TV and using computers is considered an important factor affecting PA behavior [54]. In alignment with the results of the current study regarding the level of PA before and during the lockdown as relevant to the time spent on watching TV and using the computer, another study has reported similar results [55]. The results indicated that there was a significant effect of time spent watching TV on PA before and during the lockdown. The inactive group with the time spent watching TV reported an average increase of 6.4% during the lockdown compared to the same group before the lockdown. The results of this study confirm earlier findings from another study that reported TV watching to be associated with less engagement in physical activities [56]. For example, the inactive participants with the time spent watching TV reported an average increase of 6.4% during the lockdown compared to the same group before the lockdown. Conversely, the moderate and highly active groups reported a 4% decrease during the lockdown compared to the same group before the lockdown. Similarly, another study found that 50% of participants increased their screen time, and nearly 55% decreased their PA [57]. These results are of significant importance because the findings of Werneck revealed that the combination of an increase in physical inactivity and an increase in passive screen use was associated with poor mental health during the pandemic [58].

It is very important to mention that we cannot be very negative and think about screen watching as a disadvantage because the participants in the current study might be watching the screen to keep up with the news related to COVID-19. Therefore, screen watching is considered to be crucial to keep people aware and informed on important issues related to this epidemic and for entertainment reasons. This has been highlighted by a review supporting the importance of media in disseminating information during the pandemic [59]. However, this result might suggest that screen activity should have been used to enhance health promotion and activity behaviors using proper messages and programs related to health and increasing activity level as well as its importance in disease prevention and increasing immunity.

When interpreting the results, it is important to take into account the study’s strengths and limitations. One advantage of the study is the large sample size from a diverse group of participants in the 10 Arab countries. It allows us to understand better how emerging adults with diverse ethnic/racial identities have modified their PA behavior during the pandemic. To our knowledge, this is the first study to quantify changes in PA and screen time in several Arab countries. The findings can be extended to the whole population in the 10 Arab countries. Besides the strengths of this study, it is not without limitations. The most important limitation of this study was the instrument and the data collection procedure. One of the limitations is related to the data collection before and during COVID-19 based on self-report, which may have resulted in a level of participant reporting and recalling bias. Another limitation is related to the comparison of the current study with other similar studies using different methods to measure the PA and assess the screen time, making a direct comparison of respective results and subsequent extraction of conclusions difficult.

## 5. Conclusions

In conclusion, this study showed a negative effect of lockdown on PA levels. The results of the current study suggest that understanding how PA behaviors changed during the lockdown of the COVID-19 pandemic is important to assist the health authorities in the 10 Arab countries included in the study to restructure and design new health promotion interventions in the Arab countries, including the need to focus on increasing PA levels in inactive individuals. This is very crucial to improve the health of people. Given that the results of the current study and other studies reported a decrease in PA with a concurrent increase in inactivity during the lockdown and the expected impact of these on health, it is recommended that interventions or policies be implemented to increase PA and decrease the inactivity. Examples of measures to improve the PA are bodyweight at-home workouts, online exercise classes, walking, running, and cycling outdoors. On the other hand, efforts to decrease inactivity could focus on activities such as using a standing desk and taking regular breaks from sitting.

Preventive measures of lockdown to prevent disease spreading decreased the PA of people in the Arab countries which could influence the overall health status. Health-promoting measures directed to schools, teachers, and parents in the Arab counties included the need to focus on increasing PA levels in inactive individuals, developing new policies and health promotion activities focusing on online services and messages were highly recommended. Further research on the influence of lack of PA on the overall health status as well as on the COVID-19 disease effect and duration among the Arab countries is recommended. Additionally, considering the fact that screen time is a complex issue, there is a need to undertake more research on how this affects the long-term impact of PA and sleep.

The findings support the implementation of the Global Action Plan on Physical Activity 2018–2030 and the adapted version in the Eastern Mediterranean Region includes a set of specific policy actions to guide countries in the Arab Region to accelerate and scale activities towards achieving increased levels of physical activity.

## Figures and Tables

**Table 1 ijerph-19-10832-t001:** Participants’ distribution by gender, age group, and country.

	Female		Male		Total	
	*n*	%	*n*	%	*n*	%
**Age Group**						
18–19	1082	10.9	230	9.3	1312	10.6
20–29	5126	51.5	1048	42.2	6174	49.7
30–39	1783	17.9	469	18.9	2252	18.1
40–49	1208	12.1	389	15.7	1597	12.8
50+	752	7.6	346	13.9	1098	8.8
**Country**						
Bahrain	525	5.3	120	4.8	645	5.2
Egypt	510	5.1	160	6.4	670	5.4
Jordan	1982	19.9	557	22.4	2539	20.4
Kuwait	507	5.1	143	5.8	650	5.2
Lebanon	1706	17.1	409	16.5	2115	17.0
Oman	138	1.4	26	1.0	164	1.3
Qatar	454	4.6	120	4.8	574	4.6
Saudi Arabia	2241	22.5	500	20.1	2741	22.0
Emirate	1253	12.6	287	11.6	1540	12.4
Palestine	635	6.4	160	6.4	795	6.4

**Table 2 ijerph-19-10832-t002:** The percentages and the mean and standard deviation of PA levels by country before and during lockdown.

Country	PA Before Lockdown									PA During Lockdown								
	Inactive		Low Active		Moderate Active		High Active		M ± SD	Inactive		Low Active		Moderate Active		High Active		M ± SD
	*n*	%	*n*	%	*n*	%	*n*	%		*n*	%	*n*	%	*n*	%	*n*	%	
Bahrain	197	30.5	144	22.3	213	33.0	91	14.1	1.31 ± 1.05	235	36.4	163	25.3	175	27.1	72	11.2	1.13 ± 1.03
Egypt	200	29.9	162	24.2	221	33.0	87	13.0	1.29 ± 1.03	288	43.0	156	23.3	188	28.1	38	5.7	0.96 ± 0.97
Jordan	830	32.7	623	24.5	805	31.7	281	11.1	1.21 ± 1.02	1008	39.7	638	25.1	718	28.3	175	6.9	1.02 ± 0.98
Kuwait	160	24.6	127	19.5	235	36.2	128	19.7	1.51 ± 1.07	229	35.2	123	18.9	193	29.7	105	16.2	1.27 ± 1.11
Lebanon	596	28.2	506	23.9	717	33.9	296	14.0	1.34 ± 1.03	707	33.4	557	26.3	665	31.4	186	8.8	1.16 ± 0.99
Oman	34	20.7	37	22.6	63	38.4	30	18.3	1.54 ± 1.02	46	28.0	49	29.9	50	30.5	19	11.6	1.26 ± 0.99
Qatar	133	23.2	141	24.6	215	37.5	85	14.8	1.44 ± 1.00	194	33.8	174	30.3	139	24.2	67	11.7	1.14 ± 1.01
Saudi Arabia	840	30.6	656	23.9	885	32.3	360	13.1	1.28 ± 1.04	996	36.3	630	23.0	801	29.2	314	11.5	1.16 ± 1.04
Emirate	359	23.3	355	23.1	583	37.9	243	15.8	1.46 ± 1.02	491	31.9	368	23.9	528	34.3	153	9.9	1.22 ± 1.01
Palestine	293	36.9	229	28.8	191	24.0	82	10.3	1.08 ± 1.01	342	43.0	232	29.2	157	19.7	64	8.1	0.93 ± 0.97

PA = physical activity; M = mean; SD = standard deviation.

**Table 3 ijerph-19-10832-t003:** The percentages of PA levels by selected demographic characteristics before and during lockdown.

	PA before Lockdown									PA during Lockdown								
	Inactive		Low Active		Moderate Active		High Active		F	Inactive		Low Active		Moderate Active		High Active		F
	*n*	%	*n*	%	*n*	%	*n*	%		*n*	%	*n*	%	*n*	%	*n*	%	
**Gender**																		
Female	3052	30.7	2375	23.9	3250	32.7	1274	12.8	** 53.4	3735	37.5	2486	25.0	2838	28.5	892	9.0	** 32.9
Male	590	23.8	605	24.4	878	35.4	409	16.5		801	32.3	604	24.3	776	31.3	301	12.1	
**Education**																		
≤High school	993	31.1	769	24.1	977	30.6	455	14.2	14.3	1163	36.4	785	24.6	923	28.9	323	10.1	** 9.3
Bachelor’s	2208	30.0	1746	23.7	2460	33.4	947	12.9		2748	37.3	1832	24.9	2135	29.0	646	8.8	
Graduate	441	23.5	465	24.8	691	36.8	281	15.0		625	33.3	473	25.2	556	29.6	224	11.9	
**Work Status BLD**																		
Do not work	2234	29.6	1814	24.1	2502	33.2	986	13.1	0.024	2760	36.6	1905	25.3	2202	29.2	669	8.9	** 6.3
Work	1408	28.8	1166	23.8	1626	33.2	697	14.2		1776	36.3	1185	24.2	1412	28.8	524	10.7	
**Work Status DLD**																		
Do not work	2572	29.9	2052	23.9	2829	32.9	1142	13.3	0.514	3169	36.9	2128	24.8	2495	29.0	803	9.3	0.2
Work	1070	27.9	928	24.2	1299	33.8	541	14.1		1367	35.6	962	25.1	1119	29.2	390	10.2	
**Age**																		
18–19	333	25.4	330	25.2	454	34.6	195	14.9	** 25.9	400	30.5	339	25.8	446	34.0	127	9.7	
20–29	1710	27.7	1461	23.7	2143	34.7	860	13.9		2143	34.7	1546	25.0	1920	31.1	565	9.2	** 29.5
30–39	705	31.3	548	24.3	719	31.9	280	12.4		883	39.2	567	25.2	568	25.2	234	10.4	
40–49	540	33.8	384	24.0	478	29.9	195	12.2		661	41.4	377	23.6	409	25.6	150	9.4	
50+	354	32.2	257	23.4	334	30.4	153	13.9		449	40.9	261	23.8	271	24.7	117	10.7	

PA = physical activity; BLD = before lockdown; DLD = during lockdown; F = ** 15.9.

**Table 4 ijerph-19-10832-t004:** The percentages of PA levels by time spent on watching TV and using the computer.

	PA before Lockdown									PA during Lockdown								
	Inactive		Low Active		Moderate Active		High Active		F	Inactive		Low Active		Moderate Active		High Active		F
	*n*	%	*n*	%	*n*	%	*n*	%		*n*	%	*n*	%	*n*	%	*n*	%	
**Watch TV BLD**																		
<1 h	1393	32.2	1031	23.8	1295	29.9	612	14.1	** 9.6	1733	40.0	1063	24.5	1110	25.6	425	9.8	** 8.7
1–2 h	1321	26.1	1272	25.1	1830	36.1	648	12.8		1674	33.0	1352	26.7	1608	31.7	437	8.6	
3–4 h	610	27.4	503	22.6	796	35.7	321	14.4		786	35.2	508	22.8	703	31.5	233	10.4	
5+ h	318	39.7	174	21.7	207	25.8	102	12.7		343	42.8	167	20.8	193	24.1	98	12.2	
**Watch TV DLD**																		
<1 h	928	34.3	653	24.1	774	28.6	350	12.9	** 8.4	1088	40.2	658	24.3	691	25.5	268	9.9	** 13.8
1–2 h	945	27.7	866	25.4	1200	35.2	395	11.6		1144	33.6	888	26.1	1029	30.2	345	10.1	
3–4 h	917	26.0	841	23.9	1276	36.2	489	13.9		1147	32.6	891	25.3	1167	33.1	318	9.0	
5+ h	852	30.4	620	22.2	878	31.4	449	16.0		1157	41.3	653	23.3	727	26.0	262	9.4	
**Use Computer BLD**																		
<1 h	850	35.1	582	24.0	677	28.0	312	12.9	** 14.5	981	40.5	620	25.6	594	24.5	226	9.3	** 8.4
1–2 h	1049	24.9	1053	25.0	1530	36.3	582	13.8		1346	31.9	1124	26.7	1348	32.0	396	9.4	
3–4 h	880	25.8	805	23.6	1258	36.9	468	13.7		1163	34.1	834	24.5	1083	31.8	331	9.7	
5+ h	863	36.2	540	22.6	663	27.8	321	13.4		1046	43.8	512	21.4	589	24.7	240	10.1	
**Use Computer DLD**																		
<1 h	608	36.9	405	24.6	445	27.0	190	11.5	** 6.1	698	42.4	412	25.0	392	23.8	146	8.9	** 18.0
1–2 h	546	23.7	606	26.3	866	37.6	286	12.4		662	28.7	624	27.1	784	34.0	234	10.2	
3–4 h	870	25.8	799	23.7	1236	36.6	470	13.9		1081	32.0	853	25.3	1110	32.9	331	9.8	
5+ h	1618	31.7	1170	22.9	1581	31.0	737	14.4		2095	41.0	1201	23.5	1328	26.0	482	9.4	

PA = physical activity; BLD = before lockdown; DLD = during lockdown; F = ** 15.9.

**Table 5 ijerph-19-10832-t005:** Multiple regression analysis for physical activities before and during the lockdown.

	PA before Lockdown					PA during Lockdown				
	Beta	*t*	*p*-Value	95% C.I.		Beta	*t*	*p*-Value	95% C.I.	
(Constant)		20.516	0.000	0.956	1.158		20.875	0.000	0.955	1.153
Country	0.004	0.491	0.624	0.000	0.000	0.005	0.563	0.574	0.000	0.000
Gender	0.069	8.039	0.000	0.134	0.220	0.060	7.016	0.000	0.109	0.194
Education level	0.046	5.235	0.000	0.047	0.103	0.033	3.712	0.000	0.025	0.080
Age group	−0.074	−8.221	0.000	−0.085	−0.052	−0.076	−8.456	0.000	−0.085	−0.053
Watch TV before lockdown	−0.027	−2.459	0.014	−0.058	−0.007	0.057	5.155	0.000	0.041	0.091
Watch TV during lockdown	0.062	5.531	0.000	0.039	0.082	−0.034	−3.036	0.002	−0.054	−0.012
Use computer before lockdown	−0.010	−0.832	0.405	−0.033	0.013	0.001	0.087	0.931	−0.022	0.024
Use computer during lockdown	0.007	0.568	0.570	−0.016	0.029	−0.029	−2.450	0.014	−0.050	−0.006

PA = physical activity; C.I. = confidence interval.

**Table 6 ijerph-19-10832-t006:** Paired sample *t*-test analysis for PA before and during the lockdown.

Paired Sample-*t*-Test	Paired Differences					*t*	Sig. (2-Tailed)
	Mean	SD	Std. Error Mean	95% C.I. of the Difference			
				Lower	Upper		
PA before lockdown–PA during lockdown	0.193	0.942	0.008	0.178	0.209	24.1	0.000
Work before–work during	0.086	0.319	0.003	0.080	0.091	31.6	0.000
Watch TV before–watch TV during	−0.548	0.874	0.007	−0.562	−0.533	−73.7	0.000
Use computer before lockdown– Use computer during	−0.489	0.860	0.007	−0.504	−0.475	−66.9	0.000

SD = standard deviation; Std. Error = standard error; C.I = confidence interval.

**Table 7 ijerph-19-10832-t007:** PA before and during the lockdown by country.

	PA before Lockdown				PA during Lockdown			
	Beta	*t*	Sig	95% C.I.	Beta	*t*	Sig	95% C.I.
Bahrain	−0.049	−1.235	0.217	(−0.112–0.026)	−0.010	−0.249	0.803	(−0.076–0.059)
Egypt	−0.050	−1.295	0.196	(−0.104–0.021)	−0.144	−3.759	0.000	(−0.170–−0.053)
Jordan	−0.091	−4.593	0.000	(−0.147–−0.059)	−0.057	−2.900	0.004	(−0.104–−0.020)
Kuwait	−0.076	−1.951	0.052	(−0.134–0.000)	−0.018	−0.464	0.643	(−0.086–0.053)
Lebanon	−0.116	−5.388	0.000	(−0.142–−0.066)	−0.070	−3.211	0.001	(−0.096–−0.023)
Oman	−0.046	−0.590	0.556	(−0.203–0.110)	0.064	0.822	0.412	(−0.089–0.216)
Qatar	−0.026	−0.629	0.529	(−0.095–0.049)	−0.010	−0.232	0.817	(−0.082–0.064)
Saudi Arabia	−0.034	−1.796	0.073	(−0.076–0.003)	−0.078	−4.085	0.000	(−0.123–−0.043)
Emirate	0.046	1.819	0.069	(−0.004–0.095)	0.005	0.215	0.829	(−0.043–0.054)
Palestine	−0.150	−4.278	0.000	(−0.200–−0.074)	−0.139	−3.948	0.000	(−0.183–−0.061)

PA = physical activity; C.I. = confidence interval.

## Data Availability

The raw data supporting the conclusions of this article will be made available by the corresponding author without undue reservation.

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
