# Peer review of "The Impact of COVID-19 on Physical (In)Activity Behavior in 10 Arab Countries"

_ijerph, 2022, doi:10.3390/ijerph191710832_

Round 1

Reviewer 1 Report

To thank the authors for the scientific work carried out with the purpose of deepening the knowledge of the levels of physical activity in situations of health risk and important social relationship measures such as the COVID-19 confinement.

Authors are invited to make the changes and improvements the manuscript indicated in review report.

Author Response

Dear Reviewer,

On behalf of my co-authors, I would like to thank you for your constructive comments and for the opportunity to revise our manuscript (ID: ijerph-1828785), entitled “ Impact of COVID-19 on Physical (In)Activity Behavior in 10 Arab Countries".

Please find a point-by-point response to the comments as outlined in the attached file.

We are very grateful for the positive and helpful suggestions and we feel that the quality of the manuscript has been significantly improved as a result. Thank you again for your consideration of our revised manuscript. We believe these findings would be of interest to readers of the journal and decision makers.

Sincerely,

Haleama

Reviewer 2 Report

First of all, I would like to highlight the originality of the study. 

The following are the comments derived from my review:

1) Throughout the manuscript, lowercase p should be used to refer to the p-value.

2) The aim of this study is to "assess the level of PA before and after confinement and to identify the factors affecting people's PA in the Middle East". 

There is some inconsistency between this aim and the results described, since Tables 2-7 present information before, during and after lockdown.  

It is considered crucial to revise the manuscript in this sense, modifying the aim or the results tables. It may also be due to an error in the translation of the text.  

3) In section 2 "materials and methods", it would be convenient to use subsections to present the information. For example: 2.1 Design an subjects, 2.2 Instruments and variables, 2.3 Procedure, 2.4 Data analysis.

The name of the subsections should be considered by the authors. 

4) Reference is made to the fact that an online validated survey has been used (line 132). Validation data or the reference of the source of information in which it was validated should be included. 

5) Section 2 of the manuscript should explain the criteria used to classify participants as inactive, low inactive, moderate active and high active.

6)  Statistical analyses are mentioned at the end of section 2. Before referring to the use of paired sample t test analysis, a previous statistical test should be included to measure the normal distribution of the variables, since the test used is parametric. If the result of the normality test obtained a "non-normal" value, another comparison test should be applied. 

7) In lines 221 and 222 mention is made of table 3, when in fact it is table 4.

8) Paragraph from lines 228-236. The p-value of the Multiple Regression analysis as a function of participants' country is higher than 0.050, as shown in Table 5, not lower as explained in that paragraph.

9) Maintain the use of "during lockdown" throughout the manuscript instead of also using "at the moment".

10) Table 2. Include total mean scores of PA levels (mean value for all countries).

11) The title of table 5 mentions "during lockdown" but there is an "after lockdown" column in the table. This seems to be somewhat incongruous. 

12) The title of table 7 mentions "after lockdown" but there is an "at the moment" column in the table. This seems to be somewhat incongruous.

13) The reference section should be adapted to the requirements of the journal.

  •  

Author Response

(The authors gave the same response as above.)

Round 2

Reviewer 1 Report

First of all, I would like to congratulate the authors for their good work in modifying the manuscript in response to each of the general and specific comments. The manuscript has been improved in terms of content, wording of the method and presentation of the results. The following are three aspects to modificated:

Instruments

They should indicate that this is an ad hoc questionnaire, created for the present study taking into account items from other questionnaires such as:

The statistical data of the Anova (lines 212 to 216) should be included in Results.

In the Discussion, the typography of quotations should be revised: the brackets in quotations [49], [53, 54], [59] should be closed.

Sincerely,

Author Response

Dear Reviewer,

I would like to thank you again for giving us the opportunity to revise our manuscript (ID: ijerph-1828785), entitled “ Impact of COVID-19 on Physical (In)Activity Behavior in 10 Arab Countries". We thank you for your suggestions and feedback, which have drastically improved the manuscript.

Please find attached the revised manuscript in accordance with your three points recommendations and a point-by-point response to the comments as outlined below.

Sincerely,

Haleama

Point 1: They should indicate that this is an ad hoc questionnaire, created for the present study taking into account items from other questionnaires such as:

Response 1: Thanks for your suggestion. The following sentence has been added under the sub-section 2.3 Measurement and Data Collection Procedure " An ad hoc questionnaire, created for the present study taking into account items from other questionnaires such as:..."

Point 2: The statistical data of the Anova (lines 212 to 216) should be included in Results.

Response 2: Thanks for this comment. The results from the statistical data of the Anova (lines 212 to 216) have been moved under the results section

Point 3: In the Discussion, the typography of quotations should be revised: the brackets in quotations [49], [53, 54], [59] should be closed.

Response 3: Thanks for your observation.  The typography of quotations have been revised and brackets for [49], [53, 54], [59] have been closed previously in the clean PDF version.
